ecology

range expansion, range loss, climate change, dispersal, persistence

**Author for correspondence:**
Jeremy T. Kerr
e-mail: jkerr@uottawa.ca

Invited review by the Canadian Society for Ecology and Evolution former President.

# Racing against change: understanding dispersal and persistence to improve species' conservation prospects

Jeremy T. Kerr

Department of Biology, University of Ottawa, Ottawa, Ontario, Canada K1N 6N5

  JTK, 0000-0002-3972-7560

Climate change is contributing to the widespread redistribution, and increasingly the loss, of species. Geographical range shifts among many species were detected rapidly after predictions of the potential importance of climate change were specified 35 years ago: species are shifting their ranges towards the poles and often to higher elevations in mountainous areas. Early tests of these predictions were largely qualitative, though extraordinarily rapid and broadly based, and statistical tests distinguishing between climate change and other global change drivers provided quantitative evidence that climate change had already begun to cause species' geographical ranges to shift. I review two mechanisms enabling this process, namely development of approaches for accounting for dispersal that contributes to range expansion, and identification of factors that alter persistence and lead to range loss. Dispersal in the context of range expansion depends on an array of processes, like population growth rates in novel environments, rates of individual species movements to new locations, and how quickly areas of climatically tolerable habitat shift. These factors can be tied together in well-understood mathematical frameworks or modelled statistically, leading to better prediction of extinction risk as climate changes. Yet, species' increasing exposures to novel climate conditions can exceed their tolerances and raise the likelihood of local extinction and consequent range losses. Such losses are the consequence of processes acting on individuals, driven by factors, such as the growing frequency and severity of extreme weather, that contribute local extinction risks for populations and species. Many mechanisms can govern how species respond to climate change, and rapid progress in global change research creates many opportunities to inform policy and improve conservation outcomes in the early stages of the sixth mass extinction.

## 1. Introduction

Human activities have caused extinction rates to rise sharply among populations and species in most regions of the world [1–3]. Land use change, land use intensification and overexploitation have decimated biological communities where such threats have been concentrated [4,5]. Climate has long been known to contribute vitally to the distribution of life on Earth [6,7]. Consequently, the rapidity of human-caused climate change has led to widespread biotic responses, which demonstrate the pervasive influences—and risks—of climate change on the life histories of species. Climate change is accelerating and its relative importance as a cause of present-day extinction is growing [8].

The link between where a species is found and prevailing climatic conditions in those areas is one of the oldest relationships that exists in ecology and evolutionary biology, predating the discovery of natural selection or even area effects [9,10]. Species' distributions depend on the array of environmental conditions they tolerate after accounting for antagonistic or beneficial biotic interactions and dispersal

limitation [11–13]. This understanding, which draws on classical approaches to describing species' niches [14], led to early predictions that anthropogenic climate change would cause species' geographical ranges to shift [15]. There are many precedents for such shifts throughout the complex climatic history of the Earth, with past climate changes causing redistributions of species across broad areas [16], such as those detected through ancient DNA or fossil pollen records [16,17]. Although climate sensitivities to growth in atmospheric $CO_2$ concentrations were predicted (and broadly acknowledged) in the nineteenth century [18], and confirmed and refined subsequently [19], the rapidity of both warming and the global torrent of biotic responses to it has been striking.

Range shifts are one of the clearest signs of biotic responses to climate change. They require at least one of two processes to operate: dispersal from a leading range boundary that establishes a population in an area the species did not occupy historically, or the loss of populations from historically occupied areas, such as from a trailing range boundary [20]. Leaving aside assumptions about other vital processes, like local population dynamics or adaptation, climate change will benefit a species if the rate of expansion into new areas exceeds the rate of loss elsewhere (i.e. geographical ranges grow larger). Conversely, species' extinction risks rise if range losses in some areas exceed range expansions in others (i.e. ranges get smaller), as geographical range size is strongly linked to species' extinction risk [21].

The establishment of populations in new areas is limited by, among other things, species' dispersal and persistence capacities in particular environments over both short and longer time frames [22,23]. Both processes—dispersal and persistence—are vital to understanding range dynamics during climate change. Dispersal in this context is specifically the movement of individuals of a species into an area that was previously unoccupied (i.e. dispersal that directly relates to range shift), regardless of the life stage for their dispersal. Persistence is the degree to which individuals or populations of a species within its range remain present over time. Such species responses are not fixed and can respond to selection, accelerating eco-evolutionary dynamics [24] and changes in species' traits, such as dispersal capacity in different areas of their ranges [25,26]. Other mechanisms, such as the constellation of existing and potential biotic interactions, each represent an additional challenge in terms of understanding and predicting when species will maintain or grow their geographical ranges in response to climate change [22].

Here, I review recent developments around measurements and applications of two mechanisms necessary for understanding and predicting species' range shifts during climate change. These mechanisms are (i) species' capacities to disperse to (and establish) in novel environments and (ii) how species' exposure and sensitivity (or susceptibility; [27]) to emerging conditions can be detected and used to understand their persistence in areas undergoing climate change. Emerging computational and data gathering tools are reviewed here and have enabled measurement and prediction of dispersal and persistence using datasets that are unprecedented in their spatial and temporal extents. The result has been rapid progress. Early models measured success if they detected species' range shifts in directions that were qualitatively consistent with climate change effects, while new research links species' extinction–colonization dynamics to highly resolved measurements of climate change and consequent short-term environmental variability. These developments lead to practical policy advice that could alter the trajectory of extinction rates.

## 2. Geographical range shifts and climate change

Informed by extensive understanding of how changes in palaeoclimates affected species distributions and interactions, Peters & Darling [15] wrote presciently about the prospects that anthropogenic climate change would accelerate extinction rates and cause a widespread redistribution of species. This work anticipated the need to account for interactions between land uses and climate change, the potential requirements for managed relocation of species, the problems of protecting species in nature reserves whose boundaries were fixed in place while species' range boundaries became dynamic (figure 1a), potential interactions between land use change and climate change, and the myriad challenges of accounting for changes to biotic interactions. Moreover, many species' dispersal capacities were thought to be far below rates required to track shifting climatic conditions, even in the absence of widespread habitat losses and fragmentation [15]. This work anticipated that extinction risks would rise as a consequence of climate change through impacts on species' range dynamics.

Qualitative evidence that anthropogenic climate change was beginning to take a biotic toll emerged from work comparing the limits of species' present-day geographical ranges against historical observations and asking a simple, but powerful, question: what changed [28,29]? In a series of re-surveys of known populations of the bay checkerspot butterfly (*Euphydryas editha*), population extinctions were found to be most likely in the south [30] (figure 1b). Areas that had been strongly altered through land use changes were omitted from the study, leaving climate change prominent among potential causes for this pattern. Colonization of new areas was not measured because of risks that historical surveys might not have detected all populations. This work set a precedent for using butterfly populations as a kind of sentinel taxon for the biotic effects of climate change [31]. Its focus on patterns of population extinction also differed from later work, which often emphasized more strongly the detection of range expansions along poleward range margins.

Inferences that climate change contributed to species' modern geographical range shifts grew rapidly stronger. Evidence of poleward range expansion came from a variety of taxa, though the geographical foci of early studies were most commonly European and often British. Poleward range shifts were detected among European butterfly species [28], and plant populations along elevational gradients [32,33] were found to be creeping upward altitudinally. Southern Finnish birds frequently expanded their ranges northward [34], and observations of British species across an array of taxa [35]—including several invertebrate and vertebrate groups—suggested similar trends. Such observations have expanded in both their geographical and taxonomic scopes [36]. Evidence from across many taxa, including birds, mammals, insects and plants, has led to upward adjustments in the rate at which shifts are occurring, to about 17 km decade$^{-1}$ [37]. Early estimates were rarely spatially explicit except in the directional sense: range shifts assessed along latitudinal and altitudinal gradients indicated that shifts were broadly poleward and upward, respectively.

Proc. R. Soc. B **287**: 20202061

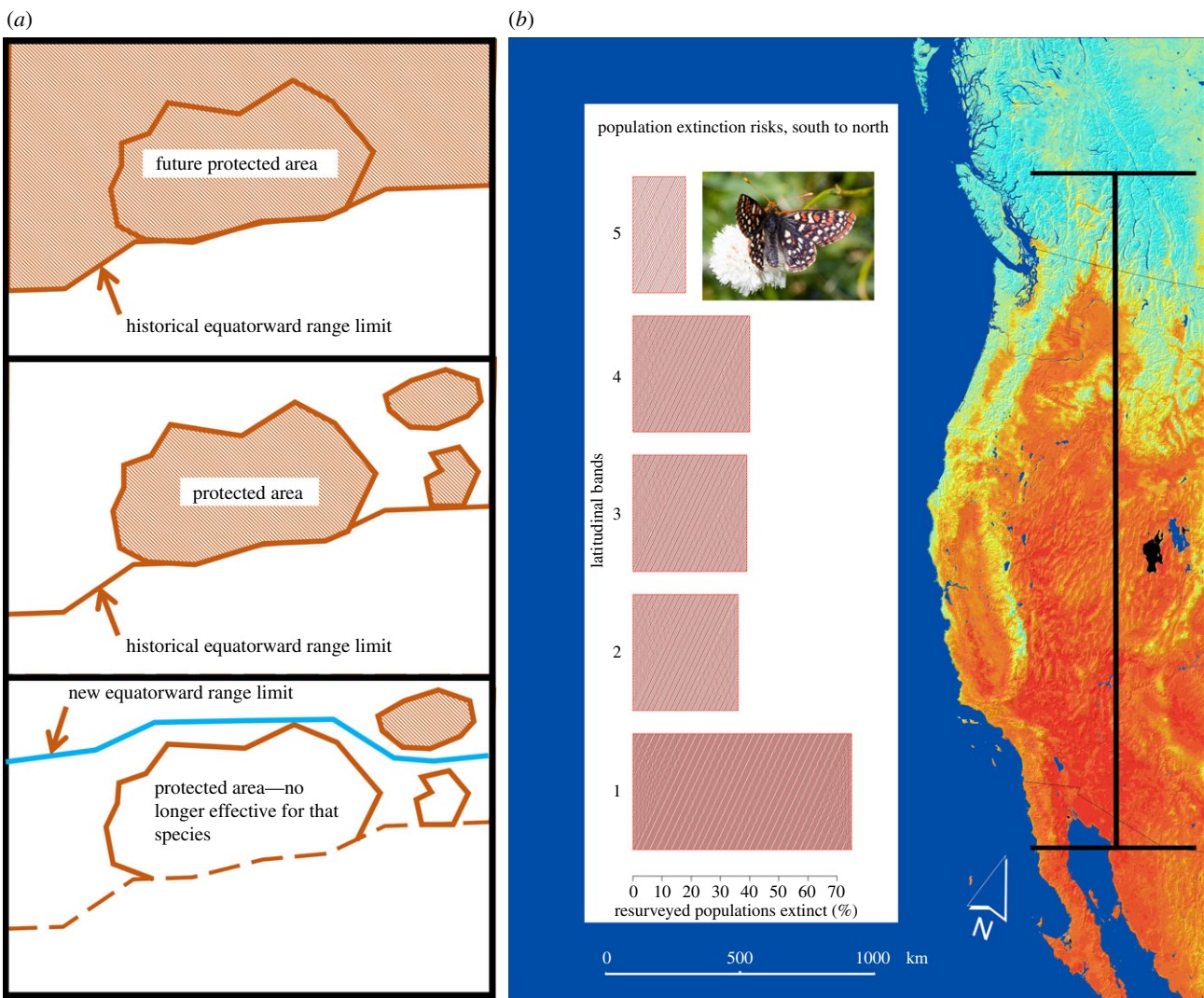

**Figure 1.** Development of critical predictions around how species might respond to climate change was followed rapidly by detection of south–north trends in extinction risk in a western butterfly species. (*a*) A species' equatorward range boundary prior to human-caused habitat losses (top panel), its range after land use change, limited largely to protected areas (middle panel), and range losses consequent to climate change that caused this species to become extinct in one of the protected areas (adapted from [15]). (*b*) Latitudinal differences in observed population extinctions of the bay checkerspot butterfly (*Euphydryas bayensis*; inset photo by Walter Sigmund, licence CC BY-SA 3.0), with the likelihood of a population extinction shown by the length of bars, on the left. The latitudinal range of population resurveys is shown in black and corresponds to the edges of the bar graph. June 2018 temperatures vary from warm (in the south) to cool (in the north) based on brightness temperature measurements from the Moderate Resolution Imaging Spectroradiometer (MODIS) sensor on Terra. These temperature data were rendered slightly transparent and overlaid on a shaded relief map to depict topographic variation also. (Online version in colour.)

The detection of such shifts and even their attribution specifically to climate change is still a vital step removed from establishing quantitative relationships with range dynamics or understanding the mechanisms underlying such relationships [38]. These mechanisms were largely and necessarily implicit in pioneering detection and attribution studies, drawing on realized niche concepts, long-term monitoring datasets, and extensive natural history expertise [12].

Measurements indicating that species were responding in various ways to anthropogenic climate change enabled correlative or other statistical tests of links between those responses and particular global change drivers [39]. That species are shifting poleward, for example, can as readily be a consequence of land use change as of warming conditions, and distinguishing between their relative roles is obviously necessary to understand whether their effects might benefit or harm species' survival prospects [40]. For example, warming could facilitate poleward expansion for a species but so could land use changes that make new habitats suitable for potential range

expansion, as has been observed among many generalist species [41]. Similarly, baseline forest cover and forest cover change, as well as climate change [42], have contributed to species redistributions along elevation gradients (based on 1464 separate observations of elevation shift from 46 locations). Among butterflies in the Sierra Nevada mountains, climate changes and habitat losses led to opposite responses, with range loss predominating in low elevation areas where habitat loss was extensive, and range expansion being more common at high elevations, where climate changes predominated [43]. Other kinds of biotic responses, such as phenological shifts, can also reflect signals of both climate and land use changes, respectively [44].

Statistical approaches provide, minimally, tests of the strength of the potential climate change signal in biological change data and can test alternative causes of those biotic responses [45]. They have the potential to be spatially explicit, testing or predicting specific trends that are particular to localities within the total study area, given a range of measured

environmental changes in those areas. These models can specify mechanisms underlying particular hypotheses, creating a pseudo-experimental framework for testing the strengths of different global change drivers [46]. Distinguishing the relative roles of climate and land use change, respectively, is difficult without more sophisticated models incorporating mechanisms that logically govern range shifts, such as dispersal or tolerance to environmental changes. Nevertheless, quantitative links between warming and range shifts are now widely documented: species in terrestrial, freshwater and marine ecosystems are shifting their geographical distributions in relation to rapid warming [47].

## 3. Dispersal and range dynamics

In the context of geographical range shifts, dispersal results in the establishment of a species' population in an area that was previously unoccupied [48]. Successful dispersal consequently leads to local expansion of a species' geographical range. Despite this vital role in global change responses, challenges associated with measuring dispersal—which varies contextually [23], potentially in response to selection [49] and interspecifically [50]—have slowed the development of models that assess its effects in specific terms. Dispersal has often been ignored entirely [51], for lack of data or because its inclusion complicates the modelling process, in predictions of species' movements in response to climate change. If the limits of individual movement are known, they can be included in models predicting shifts in species' geographical ranges over time, such as through the use of dispersal kernels (but see [52,53]). Models incorporating different dispersal scenarios are uncommon [53]. Dispersal that leads to range shifts can be modelled using species' known movement limits. Across a range of dispersal capacities (including known dispersal limits, zero dispersal and unlimited dispersal, respectively) and climate scenarios, many bumblebee species in North America are confronted with steep declines in the habitable area [53]. For some species, the disappearance of existing combinations of climatic conditions [54], not just shifts in climate zones, makes range losses inevitable [55]. Yet, dispersal is inextricably linked with many species' conservation prospects during environmental change.

Frameworks for assessing dispersal rates predate nearly all global change biology research but have been adapted to enable predictions for species' range shifts during global change. Skellam [56] developed the reaction–diffusion equation to account for random dispersal of individuals from a point of origin, enabling the predictions of changes in population density spatially and temporally. Interestingly, a motivation for this work was to explain rates of climate change-driven recolonization of Britain by oaks, which appeared to have occurred too rapidly for oak dispersal rates to permit. The mechanisms in this model—including dispersal, population growth and spatial displacement of climate zones—create a constellation of subsequent, specific predictions that help inform conservation applications [57]. Applying this framework to poleward range expansion in response to climate change (reviewed in [58]), the rate of change of species' population density, $u$, depends on the diffusion rate, $D$, of individuals through space, $x$, from occupied areas:

$$\frac{\partial}{\partial t}u = D\frac{\partial^2}{\partial x^2}u + ru, \tag{3.1}$$

where $r$ is the per capita growth rate of the dispersing species when its population size is small.

The minimum critical patch size ($L_c$) for a species in a bounded space is

$$L_c = \pi\sqrt{\frac{D}{r}}. \tag{3.2}$$

This model was adapted to test whether a species' capacity to disperse into new areas is likely to track shifting climatic conditions ($q$, the rate at which climatically suitable areas shift in space) rapidly enough to enable it to persist, assuming habitat suitability is binary:

$$L_c(q) = \pi\sqrt{\frac{D}{r}}\left(\sqrt{1 - \frac{q}{2\sqrt{Dr}}}\right)^{-1}. \tag{3.3}$$

Finally, this model predicts species persistence if

$$q < c = 2\sqrt{Dr}, \tag{3.4}$$

where $c$ is the species' capacity for dispersal and $q$ is the rate of movement of climatically tolerable conditions for the species. The critical dispersal rate to enable successful tracking of shifting climatic conditions, $D_c$, can then be calculated by rearranging the following equation:

$$D_c = \frac{q^2}{4r}. \tag{3.5}$$

This mechanistic framework predicts how quickly species' geographical ranges could shift in response to climate change, given measurements for $q$ and $r$. The rate of expected geographical displacement for species' climatic envelopes can be estimated using ubiquitous species distribution modelling methods over successive time periods. Predicted rates of range shift (figure 2a) can then be compared with habitat availability through time and species' measured dispersal rates (equations (3.3) and (3.4); see figure 2b for an example of a modelled species) to help understand extinction risk [57,58].

The existence of such frameworks that demonstrate the importance of dispersal does not imply an understanding of when dispersal rates will suffice to enable particular species to track shifting climatic conditions. A constellation of other factors can strongly affect dispersal leading to range shifts, notably interspecific (or other biotic) interactions, such as competition [59]. Whatever the mechanisms involved, dispersal 'succeeds' with the establishment of a new population in a previously unoccupied location (for a review of patterns and processes related to species' ranges, see [60]). This outcome can be broken down into two processes: movement of one or more individuals from an established location to an unoccupied one and population growth in the new location that leads to successful range extension.

There is considerable uncertainty in forecasts of both dispersal and population growth, highlighting some of the critical gaps between elegant mathematical models and messy empirical realities [61]. For example, population growth can vary stochastically both intrinsically and owing to external environmental factors, including complexes of biotic interactions [59,62]. This challenge is exacerbated because climate change is not merely the accelerated change of mean climatic conditions but is also associated with increasing variability in temperature and precipitation [63]. This variability can lead to different effects on population growth rates. For zooplankton in lakes,

to new environments, time periods or species [66,67]. Approaches to assessing the adequacy of ecological models have been summarized elsewhere, and model adequacy needs to be evaluated in light of model purpose [68].

Research into dispersal and how climatic conditions affect habitat suitability contributed to the development of conservation applications around climate connectivity to facilitate species' geographical range shifts. Four groups of approaches to this research area have been identified [69], focusing on: projected ranges for species in the future (predominantly based on species distribution models), the related question of the spatial trajectory for how climatic 'zones' will shift, existing environmental gradients that might account for directionality of species' range shifts, and the distribution of landforms or physiographic features that are associated with the eco-evolutionary processes that contribute to the origins and maintenance of biological diversity in the first place [70–72]. Accounting for differences in species' dispersal capacities informs conservation interventions, which can range from nothing at all (for strong dispersers with rapid population growth) to managed relocation (for the poorest dispersers, which may be unable to track anthropogenic climate change even in the most connected landscapes) [73]. Most species likely fall between these extremes and emerging techniques to account for their dispersal capacities and habitat requirements lead to specific recommendations for protecting particular habitat patches and corridors (despite imperfect data) to facilitate species movements through fragmented, intensively used landscapes [74,75]. Linking correlative models, like most species distribution models, with mechanism- or process-driven approaches offers potential insight that could vitally inform those planning processes [61].

The balance between species' capacities to disperse to novel environments and to maintain populations in areas that have been colonized is vital for understanding how their extinction risk changes over time [8]. Many species disperse weakly, limiting potential range expansion [76], and making it more likely that they will not track shifting climatic conditions or maintain their range size. The spatial difference between how far a species needs to move to track shifting climatic conditions and how far its range has actually shifted is its climate debt. Despite evidence of fairly rapid poleward range expansions, birds and butterflies in Europe have accumulated climate debts of 212 and 135 km, respectively [77]. Population extinctions in areas that have warmed beyond species' upper thermal limits (measured in terms of their realized niches) have been less commonly detected than range expansions, but such populations face greater extinction risks in the future.

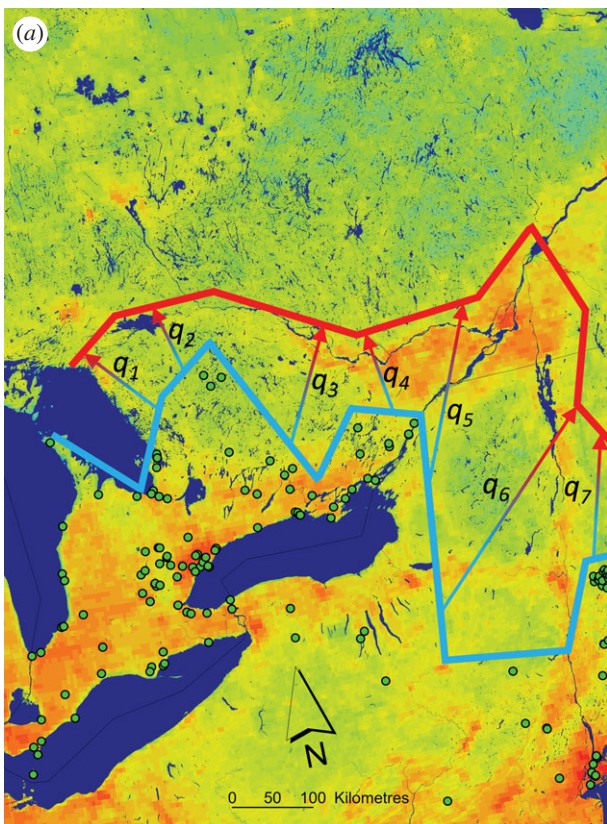

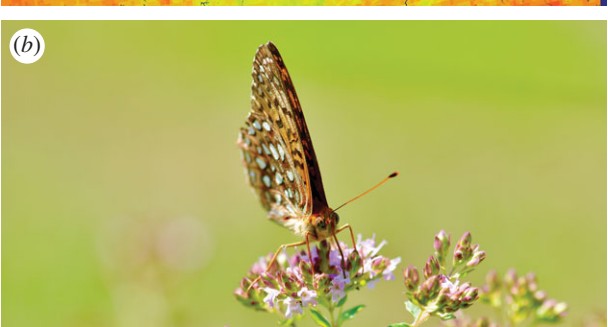

**Figure 2.** (*a*) Species distribution models or other approaches can yield estimates of the displacement of a species' geographical range over time. The gap between historical northern range margins, shown approximately as the southern line across the mapped area near the observation points (black-outlined dots) for the species, and the expected range boundary (northern line) following climate change can be measured based on the average distance between those boundaries, shown as $q_{1-7}$. This value can be compared against known dispersal capacities to predict whether the species will persist using the framework developed through equations (3.1)–(3.5). (*b*) A great spangled fritillary (*Speyeria cybele*), one of many butterfly species whose poleward range expansion in response to climate change has been modelled in North America. Photo by J. Kerr. (Online version in colour.)

growing temperature variability associated with climate could lead to higher overall population growth rates [64]. In a separate study of 644 paired population growth measurements from high and low levels of biotic, abiotic and human-related drivers, population growth proved to be highly sensitive to each class of driver [65]. The potential complexity of models needed to predict these aspects of species' range dynamics is very high: models could include many imperfectly measured mechanisms that interact transiently or stably, operate differently over short- and long time periods, and vary in their strengths along spatial and environmental gradients. Models with many moving parts are hard to parameterize and harder still to validate or transfer

## 4. Persistence during climate change

Extinctions have been documented among populations present in areas where warming now exceeds the upper thermal limits of species' realized niches [78,79]. Given species' life-history characteristics, warming has rendered populations in areas near the warm limits of species' realized niches particularly susceptible to extinction [49]. Bumblebee species' ranges across Europe and North America tended to retract from their historical southern limits relative to historical baselines, indicating losses among southern populations [79]. Bumblebees originated in temperate environments [80], and present-day trends of

population decline among species in North America and Europe from warm, southern areas have a phylogenetic signal that may reflect this taxon's evolutionary origins. In other words, niche conservatism in conjunction with temperate origins for species represents a distinct (though far from exclusive) pathway towards elevated extinction risk.

Among tropical reptile populations, extinction risks have risen as climate changes have imposed conditions at or beyond species' physiological limitations, indicating that elevated extinction risk due to climate change is not limited to lineages that originated in temperate environments. The probability of local population extinctions in these reptiles is expected to grow to about 20% by 2080 [78]. Moreover, for both bumblebees and tropical reptiles, the rates of range expansion into new areas do not come close to offsetting the rates of range loss elsewhere, indicating that climate change is tilting the balance strongly towards elevated extinction risk. After accounting for dispersal limitation and realistic extinction debt assessments, about 1 in 6 species by 2080 will face elevated extinction risks from climate change, given the present trajectory of carbon emissions [8]. While this estimate is lower than earlier projections [81], it implies rates of loss that are comparable with mass extinctions, especially if those rates are maintained or rise beyond 2080 [4,82].

'Climate chaos' is the growing frequency and intensity of extreme weather events associated with climate change [83] and it is linked to species' local extinction risk both in the fossil record and, increasingly, in relation to anthropogenic climate change. Such effects may help explain why population extinctions are sometimes more rapid than predicted given changing climatic conditions [27], though other extinction drivers, such as land use change and intensification [84], and overkill [85,86], certainly contribute strongly to population and species losses. Nevertheless, most models of climate change-driven species declines, such as obtained through species distribution modelling, assess habitat suitability using climate measurements derived from weather averages over various time periods (commonly 30 years [87]) that many organisms do not live long enough to experience. For many organisms, it is not a changing climate that directly affects them, but the fluctuations in weather associated with climate change. Consequently, it is valuable to distinguish between 'press' (longer-term) exposures to novel conditions related to climate change and 'pulse' (episodic) exposures that include weather extremes [88].

The potential effects of climate change, as mediated by changing weather, reflect organismal sensitivity to new conditions, exposure to those conditions and adaptive potential [89]. Consequently, extreme weather events, such as heatwaves, drought, or storms, contribute to an array of localized biotic responses [90]. Effects can be direct (e.g. on physiological performance) or indirect (e.g. through emergent phenological mismatches between pollinators and host plants) [91–94]. Growing frequencies and intensities of such extreme events can be rapid and catastrophic, such as widely observed degradation of tropical coral reef communities, which are highly susceptible to warming water temperature associated with climate change [95], despite the smaller role of some stressors, like habitat fragmentation, that pervasively complicate and inhibit the movement and persistence of individuals in terrestrial habitats.

There can be strong interplay between dispersal and local persistence due to extreme weather. In terrestrial systems, habitat corridors between species' current ranges and areas to which they must move are likely to be more effective if they provide microclimatic refugia—limiting individual or small population exposure to extreme events and to shifts in mean conditions (e.g. warming average temperatures) [96,97]. Among studies examining biotic effects of extreme weather, 57% (of 534 studies found in 205 journals, between 1941 and 2015) reported negative impacts and nearly a fifth found that populations declined by 25% or more [98]. In these studies, the rates of recovery following extreme weather events (e.g. heatwaves, storms or drought) varied, with 30% of studies showing that populations did not recover and most others showing recoveries taking 2–10 years. For many other taxonomic groups, exposure to 'pulse' conditions (namely, extreme events) represents a key pathway for biological impacts that is distinct from long-term 'press' exposures.

Species' sensitivities to new weather conditions should reflect the proximity of their realized niche boundaries (e.g. their upper thermal limits), while exposure depends on varying magnitudes of weather changes throughout species' ranges relative to their tolerance limits. Both sensitivity and exposure have previously been linked to climate change-driven extinction risks globally, as measured using expert opinion, and both will vary spatially [27,89]. For a hypothesized environmental limitation on the distribution and abundance of a species over a broad area, the species' proximity to its tolerance limits (sensitivity) relative to changing conditions (exposure) can be estimated as [99]

$$P = \frac{1}{t} \sum_{i=1}^{t} \left( \frac{N_\mathrm{m} - N_\mathrm{Smin}}{N_\mathrm{Smax} - N_\mathrm{Smin}} \right), \tag{4.1}$$

where $P$ is the species' position in realized niche space assessed based on weather data over a time period of interest divided into $t$ units (e.g. 12 if the time units are months), $N_\mathrm{m}$ is the environmental measurement for that area during that month, $N_\mathrm{Smax}$ is the species' upper realized niche limit, and $N_\mathrm{Smin}$ is the species' lower realized niche limit. For bumblebee species in Europe and North America, a measurement of thermal position between a baseline and recent time period (1901–1974 and 2000–2014) explains spatio-temporal range dynamics, including extinction and colonization, better than temperature or precipitation change (electronic supplementary material, figure S1). This measurement is easily implemented in geographic information systems and can be calculated readily across broad areas and through time using weather data. It can also be integrated across species assemblages, as with climate metrics such as the community temperature index [77], to help predict geographical variation in extinction risks, or potentially to inform the development of corridors, stepping stone habitats, or protected area management strategies. Changing frequencies and intensities of extreme weather conditions were related to the likelihood of population persistence and colonization of new areas, spatially and temporally.

## 5. Mechanisms, models and management

The challenge is to produce models that successfully predict whether, where and when species will decline or benefit from climate change and other forms of global change [45,100]. To achieve this, models 'should be made as simple as possible, but no simpler'[1] (see also [61]). At one extreme, it is possible

to construct correlative models describing a species' distribution on the basis of an array of environmental conditions, such as those derived through species distribution models [101]. Such a model can capture real biological processes that generate measurable patterns that can improve predictions of global change impacts on species, such as their dependence on temperature in aspects of life history. Conversely, these models can sometimes describe distributions purely because species' ranges are commonly continuous through space and environmental factors can have a similar spatial structure to species' ranges [102,103]. That causation is hard to infer from pure correlation is axiomatic, necessitating strategies to avoid weak inference [104], such as assessing spatial models' transferability through time [105] or to independent locations [106]. At the other extreme, mechanisms affecting how species respond to climate change have been grouped into six broad categories: biotic interactions, evolutionary responses, physiology, dispersal and range dynamics, demography and life-history characteristics, and responses to environmental variation [22]. Each of these groups includes an unknown number of more specific mechanisms, many of which necessarily interact (e.g. dispersal can respond rapidly to selection, life-history characteristics like phenology interact with range dynamics, etc.). Testing each of these mechanisms thoroughly is impossible, as no measurements (let alone field-based tests) for any of them exist for most described species [22], and most species have probably not been described [107,108]. The challenge, from a conservation perspective, is to identify the mechanisms that are necessary for predicting population or species' extinction risks. This uncontroversial objective is analogous to statistical frameworks designed to balance model complexity with model information (as well as Einstein's advice, quoted above) [109,110].

Accounting for critical mechanisms, such as dispersal or the limits of species' physiological tolerances, in models of species' extinction risk is difficult but obviously necessary for many practical purposes. Models that correctly identify vital mechanisms predicting species' potential range expansions or losses are more useful because mechanisms improve prediction of biological responses to novel conditions and can consequently indicate where intervention, planning, and management are necessary [22,45]. For example, recognition of dispersal limitation in fragmented landscapes launched research programmes into the roles of habitat corridors, and recognition that species' persistence can be threatened by some kinds of extreme weather events has led to calls to account for microrefugia and topoclimatic heterogeneity in landscape management [111]. Alternatively, a purely correlative model might link the distribution of a species to an array of environmental factors and describe that species' distribution ably. Without evidence of causation, purely correlative models' main contribution is to suggest avenues for potentially useful further research. For example, correlations between numbers of species within a region and that region's mean climatic conditions are well known [6,7,112], and there is little doubt that climate is causally related to gradients of biological diversity. However, early correlative models linking regional climate to species richness were exploratory and implied little about how to conserve biological diversity potentially threatened by climate change beyond mitigating climate change itself. Recognition of potential mechanisms governing the climate–richness relationship, such as niche conservatism and evolutionary origins in tropical or temperate climates [79], suggests avenues for specific

management actions and could inform decisions about which species are vulnerable to rapid warming.

The roles of mechanisms, such as dispersal capacity, can be assessed and benefit management interventions even when such traits cannot be precisely measured in complex environments [25]. The critical requirement is that there must be strong confidence that the mechanism actually affects the likelihood of conservation success. The interaction between land use change and intensification and climate change will clearly amplify threats to biological diversity for many species, with climate change imposing a requirement to shift into new areas while land use changes create barriers to that movement [113], likely accentuating species' extinction risks [114]. Such effects are mediated by species' traits. Research into landscape connectivity predates widespread recognition of extinction risks posed by anthropogenic climate change and can be traced back, at least, to original applications of island biogeographic theory to protected area design [115]. Interactions between landscape connectivity and climate change were recognized much later, but approaches to their assessment have advanced rapidly and are beginning to inform continent-wide conservation planning research.

## 6. Conclusion

Understanding of the consequences of global climate change for biological diversity and its conservation has grown exponentially since those risks were first outlined. Whether through its direct or indirect effects on species, predictions that climate change alone is sufficient to launch a mass extinction are credible, as are observations that this process has begun. Other aspects of human activity, such as land use change or the spread of invasive species, are sufficient to create such a crisis directly, and the contributions of multiple stressors, including climate change, amplify those risks.

Geographical range shifts among species have now been documented in terrestrial, marine and freshwater ecosystems globally. Every such shift can include range expansion, achieved through dispersal to (and establishment in) new areas or range loss, the consequence of local population losses from historically occupied areas. These are necessary logical foundations for research into range dynamics, and progress towards understanding how such processes alter species' ranges has been rapid.

Mechanistic approaches to considering dispersal can yield valuable and surprising benefits, such as predictions for the minimum patch sizes needed to conserve species whose ranges are shifting. The growing realization that climate change exerts some of its most serious effects through changes in the frequency and intensity of extreme weather reflects the formal recognition that consequences to species' extinction risks reflect their sensitivities and exposure to such change, as well as their adaptive capacities. Much progress is needed to understanding whether, when and why species may adapt to or tolerate changing weather patterns associated with climate change. However, new techniques that account for both species' sensitivities and exposure to change inform predictions for colonization and extinction and consequently for range dynamics overall.

The motivations of research into range dynamics can be as diverse as the researchers who pursue them, ranging from academic curiosity to a desire to inform practical policy goals. As with conservation biology [116], however, global

change biology is at least partly a mission-driven discipline whose core objective is to minimize the negative impacts of pressures such as climate change, whether for populations, species or ecosystem processes. In terms of species conservation, rapid progress is vital and should be commensurate with the pace and global scale of climate change itself, informing practical strategies to reduce conservation threats, such as identifying networks that minimize species' exposures to intolerable conditions and maximize their capacity for movement. Understanding the critical processes and mechanisms that affect species' range dynamics is vital to identifying useable solutions. Knowledge of how and why global change affects biological diversity is constantly evolving but is more than sufficient to inform policy options that would change the trajectory of the sixth mass extinction for the better.

Data accessibility. This article does not contain any additional data.
Competing interests. I declare I have no competing interests.

Funding. I am grateful to the Natural Sciences Engineering Research Council of Canada for Discovery Accelerator Supplement and Discovery Grant funds, to MITACS for fellowship support for my research group, and for funding through the University of Ottawa Research Chair in Macroecology and Conservation.
Acknowledgements. This review is submitted as the valedictory contribution of an outgoing President of the Canadian Society for Ecology and Evolution. This work is dedicated to Evan and Elise Kerr, who I hope can grow up in a world undiminished, and to the memory of Robert M. May, through whose guidance I learned much about how science and policy must interact if urgent problems are to be solved. Discussions with Peter Soroye, Catherine Sirois-Delisle, Olga Koppel, Susan Gordon, Sarah Chisholm, Kirsten Crandall and Anouk Paradis improved this work.

## Endnote

[1]This quotation is attributed to A. Einstein, who may or may not have said it in this way, but who expressed this idea in writings in a longer form.

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
