## [Reviewer comments · Proceedings of the Royal Society B: Biological Sciences]

Review History

RSPB-2020-2061.R0 (Original submission)

Review form: Reviewer 1

Recommendation

Accept with minor revision (please list in comments)

Scientific importance: Is the manuscript an original and important contribution to its field?

Good

General interest: Is the paper of sufficient general interest?

Good

Quality of the paper: Is the overall quality of the paper suitable?

Good

Is the length of the paper justified?

No

Should the paper be seen by a specialist statistical reviewer?

No

Do you have any concerns about statistical analyses in this paper? If so, please specify them explicitly in your report.

No

It is a condition of publication that authors make their supporting data, code and materials available - either as supplementary material or hosted in an external repository. Please rate, if applicable, the supporting data on the following criteria.

Is it accessible?

N/A

Is it clear?

N/A

Is it adequate?

N/A

Do you have any ethical concerns with this paper?

No

Comments to the Author

Species range shifts are the conclusion of many ecological processes in response to global change. This review focuses on two key processes acting on range shifts: dispersal and persistence. Overall, the paper is well written and sums well the aspects that could be useful for improving conservation strategies and policies. The strength of the paper stems from the concrete examples of species range shifts, the weakness is that the presentation of models throughout the manuscript is very superficial and somehow not reflecting the state-of-the-art in the field of species distribution modeling. I would recommend the removal of some sections on models so that other sections can be updated and improved. Then, the terms “dispersal” and “persistence” should be defined early in the paper. Indeed, dispersal could be referring to natal dispersal, breeding dispersal, dispersal rate, dispersal distance, etc., and persistence is the sum of many processes. It should be stated how these two terms are used throughout the text. Below, I indicate where the text could be improved.

Abstract, lines 52-53: “...statistical tests distinguishing between climate change and other global change drivers followed quickly.” This part of the sentence is not clear. What is the take-home message here? Please clarify or remove.

Abstract, line 61: “...climatic conditions elsewhere ...”: Where is “elsewhere”? This is not clear. I would remove this part of the statement.

Line 70: “... population ...”: Several populations, right? So “populations”.

Line 102: In this paragraph, it would be the best place to define the term dispersal and persistence in more detail. Indeed, to establish in a new location, individuals of a species would need to compete for space and resources.

Lines 108-113: Either remove this section or move it in another section that presents more explicitly the limitations of models.

Line 117: “... species sensitivity ...” to be used as “indicator species”? Please clarify and add some references.

Line 119: “... these techniques...”: Sorry, which techniques? Please clarify.

Lines 120-124: This is a long sentence: Please make two or three sentences. Please clarify the last part of the sentence and add some references (“... highly resolved measurements of climate change and its consequent short term variability ...”).

Lines 136-137: “Moreover, many species’ dispersal capacities were thought to be far below rates required to track shifting climatic conditions, ...”. Please add some references.

Line 162: Please clarify for which species/taxa this shift of 17km/decade referred to.

Lines 211-212: “In some studies, dispersal has been assessed using dispersal kernels.” This sentence can be removed as it is already mentioned.

Lines 218-248: I strongly suggest to only summarize the key processes and take-home message of Leroux et al. 2103 [59] and remove the five equations and respective text. By doing so, the message could be stressing whether or not the modeling approach is applicable to all taxa or not.

Lines 249-254: It would be good to cite the book of Kevin Gaston (2003) here (and throughout the text).

Gaston KJ. 2003. The structure and dynamics of geographic ranges. Oxford University Press.

Lines 255-261: There are many recent papers that could be relevant to cite here (see below for some suggestions).

Briscoe NJ, et al. Forecasting species range dynamics with process-explicit models: matching methods to applications. *Ecology Letters*. 2019 Nov;22(11):1940-56.

Getz WM, et al. Making ecological models adequate. *Ecology letters*. 2018 Feb;21(2):153-66.

Naujokaitis-Lewis IR, et al. Uncertainties in coupled species distribution–metapopulation dynamics models for risk assessments under climate change. *Diversity and Distributions*. 2013 May;19(5-6):541-54.

Naujokaitis-Lewis I, Fortin MJ. Spatio-temporal variation of biotic factors underpins contemporary range dynamics of congeners. *Global change biology*. 2016 Mar;22(3):1201-13.

Line 319: “Climate chaos”: Any reference for this term?

Lines 330-332: Please remove or expend.

Line 350-351: Please add so concluding statement for this sentence/statement.

Line 372: Please add some references. These two are only examples of many potential references that could be used from Carlos Carroll and others.

D'Aloia CC, et al. Coupled networks of permanent protected areas and dynamic conservation areas for biodiversity conservation under climate change. *Frontiers in Ecology and Evolution*. 2019 Feb 14;7:27.

Huang JL, et al. Importance of spatio-temporal connectivity to maintain species experiencing range shifts. *Ecography*. 2020 Apr;43(4):591-603.

Line 380: Again, please cite recent reviews like Getz et al. (2018) and Briscoe et al. (2019).

Review form: Reviewer 2

Recommendation

Accept with minor revision (please list in comments)

Scientific importance: Is the manuscript an original and important contribution to its field?

Excellent

General interest: Is the paper of sufficient general interest?

Excellent

Quality of the paper: Is the overall quality of the paper suitable?

Excellent

Is the length of the paper justified?

Yes

Should the paper be seen by a specialist statistical reviewer?

No

Do you have any concerns about statistical analyses in this paper? If so, please specify them explicitly in your report.

No

It is a condition of publication that authors make their supporting data, code and materials available - either as supplementary material or hosted in an external repository. Please rate, if applicable, the supporting data on the following criteria.

Is it accessible?

N/A

Is it clear?

N/A

Is it adequate?

N/A

Do you have any ethical concerns with this paper?

No

Comments to the Author

RSPB-2020-2061

Overall comments:

This review covers a timely topic, dispersal and persistence of species in response to climate change. It is well written, and effectively incorporates both empirical observations and models, and tells a story. It will be accessible to a broad audience. It focuses on two recent developments regarding two mechanisms for predicting range shifts related to climate change: 1) dispersal and establishment, and 2) exposure and sensitivity to emerging conditions. Overall, the implications of this field could influence policy and species' persistence. The review develops early ideas of range shifts as singular measurements into being a consequence of multiple anthropogenic drivers (including land use change) that contribute to species redistributions. This then builds to the development of dispersal models in the context of shifting climatic conditions, and the application to conservation for species based on dispersal capacity. There are a range of examples included, from butterflies and birds, to bees and reptiles and more. However, more it focused on terrestrial than marine and freshwater aquatic systems, though the latter are briefly mentioned in passing. It is interesting to think about how aquatic systems deal with climate change, from drying of ponds to warming water, and that in marine systems the oceans are still contiguous unlike terrestrial habitats, therefore some different processes may be operating depending on the species and types of disturbance. The review also includes extreme climate events and its role in extinctions. It presents ongoing challenges, including formulation of models that meaningful but not overly complicated to be so specific and untenable, particularly in the context of applications for conservation.

The figures used are well done and effective in visually communicating main points of the review.

Specific comments:

L. 63-64: I don't quite follow this, as it's a jump from talking about species to individual organisms. "...which can be measured on time scales relevant for individual survival and used to inform predictions of local extinction risk."

L.227: Equation (1): does 'x' have to be defined in this model?

L.444-4455: Perhaps "...marine, and freshwater" instead of "...marine, and aquatic"

Figure 2a: Concerned that the red and green solid lines may not be visible for red/green colourblind readers, although the text does refer to the line position as well.

Decision letter (RSPB-2020-2061.R0)

05-Oct-2020

Dear Dr Kerr:

Your manuscript has now been peer reviewed, and their comments (not including confidential comments to the Editor) are included at the end of this email for your reference. As you will see, the reviewers like your review, as do I, but they have raised some concerns with your manuscript and I would like to invite you to revise your manuscript to address them.

We do not allow multiple rounds of revision so we urge you to make every effort to fully address all of the comments at this stage. If deemed necessary, your manuscript will be sent back to one or more of the original reviewers for assessment. If the original reviewers are not available we may invite new reviewers. Please note that we cannot guarantee eventual acceptance of your manuscript at this stage.

Research ethics:

Use of animals and field studies:

It is a condition of publication that you make available the data and research materials supporting the results in the article (<https://royalsociety.org/journals/authors/author-guidelines/#data>). Datasets should be deposited in an appropriate publicly available repository and details of the associated accession number, link or DOI to the datasets must be included in the Data Accessibility section of the article (<https://royalsociety.org/journals/ethics-policies/data-sharing-mining/>). Reference(s) to datasets should also be included in the reference list of the article with DOIs (where available).

Please submit a copy of your revised paper within three weeks. If we do not hear from you within this time your manuscript will be rejected. If you are unable to meet this deadline please let us know as soon as possible, as we may be able to grant a short extension.

Best wishes,
Innes Cuthill
Prof. Innes Cuthill
Reviews Editor, Proceedings B
<mailto:proceedingsb@royalsociety.org>

Reviewer(s)' Comments to Author:

Referee: 1

Comments to the Author(s)

Species range shifts are the conclusion of many ecological processes in response to global change. This review focuses on two key processes acting on range shifts: dispersal and persistence.

Overall, the paper is well written and sums well the aspects that could be useful for improving conservation strategies and policies. The strength of the paper stems from the concrete examples of species range shifts, the weakness is that the presentation of models throughout the manuscript is very superficial and somehow not reflecting the state-of-the-art in the field of species distribution modeling. I would recommend the removal of some sections on models so that other sections can be updated and improved. Then, the terms "dispersal" and "persistence" should be defined early in the paper. Indeed, dispersal could be referring to natal dispersal, breeding dispersal, dispersal rate, dispersal distance, etc., and persistence is the sum of many processes. It should be stated how these two terms are used throughout the text. Below, I indicate where the text could be improved.

Abstract, lines 52-53: "...statistical tests distinguishing between climate change and other global change drivers followed quickly." This part of the sentence is not clear. What is the take-home message here? Please clarify or remove.

Abstract, line 61: "...climatic conditions elsewhere ...": Where is "elsewhere"? This is not clear. I would remove this part of the statement.

Line 70: "... population ...": Several populations, right? So "populations".

Line 102: In this paragraph, it would be the best place to define the term dispersal and persistence in more detail. Indeed, to establish in a new location, individuals of a species would need to compete for space and resources.

Lines 108-113: Either remove this section or move it in another section that presents more explicitly the limitations of models.

Line 117: "... species sensitivity ..." to be used as "indicator species"? Please clarify and add some references.

Line 119: "... these techniques...": Sorry, which techniques? Please clarify.

Lines 120-124: This is a long sentence: Please make two or three sentences. Please clarify the last part of the sentence and add some references ("... highly resolved measurements of climate change and its consequent short term variability ...").

Lines 136-137: "Moreover, many species' dispersal capacities were thought to be far below rates required to track shifting climatic conditions, ...". Please add some references.

Line 162: Please clarify for which species/taxa this shift of 17km/decade referred to.

Lines 211-212: "In some studies, dispersal has been assessed using dispersal kernels." This sentence can be removed as it is already mentioned.

Lines 218-248: I strongly suggest to only summarize the key processes and take-home message of Leroux et al. 2103 [59] and remove the five equations and respective text. By doing so, the message could be stressing whether or not the modeling approach is applicable to all taxa or not.

Lines 249-254: It would be good to cite the book of Kevin Gaston (2003) here (and throughout the text).

Gaston KJ. 2003. The structure and dynamics of geographic ranges. Oxford University Press.

Lines 255-261: There are many recent papers that could be relevant to cite here (see below for some suggestions).

Briscoe NJ, et al. Forecasting species range dynamics with process-explicit models: matching methods to applications. *Ecology Letters*. 2019 Nov;22(11):1940-56.

Getz WM, et al. Making ecological models adequate. *Ecology letters*. 2018 Feb;21(2):153-66.

Naujokaitis-Lewis IR, et al. Uncertainties in coupled species distribution–metapopulation dynamics models for risk assessments under climate change. *Diversity and Distributions*. 2013 May;19(5-6):541-54.

Naujokaitis-Lewis I, Fortin MJ. Spatio-temporal variation of biotic factors underpins contemporary range dynamics of congeners. *Global change biology*. 2016 Mar;22(3):1201-13.

Line 319: “Climate chaos”: Any reference for this term?

Lines 330-332: Please remove or expend.

Line 350-351: Please add so concluding statement for this sentence/statement.

Line 372: Please add some references. These two are only examples of many potential references that could be used from Carlos Carroll and others.

D'Aloia CC, et al. Coupled networks of permanent protected areas and dynamic conservation areas for biodiversity conservation under climate change. *Frontiers in Ecology and Evolution*. 2019 Feb 14;7:27.

Huang JL, et al. Importance of spatio-temporal connectivity to maintain species experiencing range shifts. *Ecography*. 2020 Apr;43(4):591-603.

Line 380: Again, please cite recent reviews like Getz et al. (2018) and Briscoe et al. (2019).

Referee: 2

Comments to the Author(s)

RSPB-2020-2061

Overall comments:

This review covers a timely topic, dispersal and persistence of species in response to climate change. It is well written, and effectively incorporates both empirical observations and models, and tells a story. It will be accessible to a broad audience. It focuses on two recent developments regarding two mechanisms for predicting range shifts related to climate change: 1) dispersal and establishment, and 2) exposure and sensitivity to emerging conditions. Overall, the implications of this field could influence policy and species' persistence. The review develops early ideas of range shifts as singular measurements into being a consequence of multiple anthropogenic drivers (including land use change) that contribute to species redistributions. This then builds to the development of dispersal models in the context of shifting climatic conditions, and the application to conservation for species based on dispersal capacity. There are a range of examples included, from butterflies and birds, to bees and reptiles and more. However, more it focused on terrestrial than marine and freshwater aquatic systems, though the latter are briefly mentioned in passing. It is interesting to think about how aquatic systems deal with climate change, from drying of ponds to warming water, and that in marine systems the oceans are still contiguous unlike terrestrial habitats, therefore some different processes may be operating depending on the species and types of disturbance. The review also includes extreme climate events and its role in extinctions. It presents ongoing challenges, including formulation of models that meaningful but not overly complicated to be so specific and untenable, particularly in the context of applications for conservation.

The figures used are well done and effective in visually communicating main points of the review.

Specific comments:

L. 63-64: I don't quite follow this, as it's a jump from talking about species to individual organisms. "...which can be measured on time scales relevant for individual survival and used to inform predictions of local extinction risk."

L.227: Equation (1): does 'x' have to be defined in this model?

L.444-4455: Perhaps "...marine, and freshwater" instead of "...marine, and aquatic"

Figure 2a: Concerned that the red and green solid lines may not be visible for red/green colourblind readers, although the text does refer to the line position as well.

Author's Response to Decision Letter for (RSPB-2020-2061.R0)

See Appendix A.

Decision letter (RSPB-2020-2061.R1)

04-Nov-2020

Dear Dr Kerr

I am pleased to inform you that your manuscript entitled "Racing against change: understanding dispersal and persistence to improve species' conservation prospects*" has been accepted for publication in Proceedings B.

If you are likely to be away from e-mail contact during this period, let us know. Due to rapid publication and an extremely tight schedule, if comments are not received, we may publish the paper as it stands.

Your article has been estimated as being 12 pages long. Our Production Office will be able to confirm the exact length at proof stage.

Open access

You are invited to opt for open access via our author pays publishing model. Payment of open access fees will enable your article to be made freely available via the Royal Society website as soon as it is ready for publication. For more information about open access publishing please visit our website at http://royalsocietypublishing.org/site/authors/open_access.xhtml.

The open access fee is £1,700 per article (plus VAT for authors within the EU). If you wish to opt for open access then please let us know as soon as possible.

Paper charges

Sincerely,
Proceedings B
mailto: proceedingsb@royalsociety.org

Appendix A

uOttawa

Université d'Ottawa | University of Ottawa

Faculté des sciences | Faculty of Science
Département de biologie | Department of Biology
Pavillon Gendron | Gendron Hall
30 Marie-Curie Ottawa ON Canada K1N 6N5
☎ 613-562-5718 ☎ 613-562-5486 bio@uOttawa.ca

Tuesday, October 27, 2020

Dear Dr. Cuthill,

I appreciated the insightful commentary regarding this review paper (submitted as an outgoing President of the *Canadian Society for Ecology & Evolution*) and I have revised this work accordingly. I believe I have addressed remarks from both reviewers comprehensively and the resulting manuscript should speak clearly and to a broad audience.

Below, I include specific responses (*in italics*) to every comment from the reviewers (in bold). I have also included a fully revision-marked version of the manuscript that should make it straightforward to see areas where I have made changes. There are a small number of typographical changes also. I was uncertain whether that revision-marked text needed to be appended to this revision letter, so that is what I have done. I have submitted a “clean” version separately.

I will be pleased to address any further issues you may identify that would make this work stronger, more inclusive, or important to a broader readership.

Sincerely,

Jeremy Kerr
Biology, University of Ottawa
jkerr@uottawa.ca

uOttawa

Université d'Ottawa | University of Ottawa

Faculté des sciences | Faculty of Science
Département de biologie | Department of Biology
Pavillon Gendron | Gendron Hall
30 Marie-Curie Ottawa ON Canada K1N 6N5
☎ 613-562-5718 ☎ 613-562-5486 bio@uOttawa.ca

Referee: 1

Comments to the Author(s)

Species range shifts are the conclusion of many ecological processes in response to global change. This review focuses on two key processes acting on range shifts: dispersal and persistence. Overall, the paper is well written and sums well the aspects that could be useful for improving conservation strategies and policies.

Grateful for the kind words regarding the paper's overall approach and content.

The strength of the paper stems from the concrete examples of species range shifts, the weakness is that the presentation of models throughout the manuscript is very superficial and somehow not reflecting the state-of-the-art in the field of species distribution modeling. I would recommend the removal of some sections on models so that other sections can be updated and improved. Then, the terms “dispersal” and “persistence” should be defined early in the paper. Indeed, dispersal could be referring to natal dispersal, breeding dispersal, dispersal rate, dispersal distance, etc., and persistence is the sum of many processes. It should be stated how these two terms are used throughout the text.

I have examined places in the text where these themes can be included. The logical structure of the paper is intended to move from introduction into a recognition of the recent origins of this field, and more specific coverage around dispersal and persistence, respectively. The main focus is on discussing how these crucial mechanisms are being used to understand range dynamics. The concluding section of this work demonstrates many ways that mechanism-driven models are being applied practically to inform conservation and management decision making. A thread throughout is the tension between complex and simple models. I have included references to many papers employing species distribution models and reviewing this topic per se is not a goal of this work.

I have rebalanced these discussions in response to the reviewer's comments. I have incorporated many of the reviewer's suggested references and included explanatory text to support those references. I have expanded the focus on other approaches to evaluating dispersal in the context of geographical range shifts. I detail the specific changes and where those changes are in the text next to the specific suggestions from the reviewer.

Lines 111-114: I have added definitions for the terms dispersal and persistence in the context of geographical range shifts.

Abstract, lines 52-53: “...statistical tests distinguishing between climate change and other global change drivers followed quickly.” This part of the sentence is not clear. What is the take-home message here? Please clarify or remove.

uOttawa

Université d'Ottawa | University of Ottawa

Faculté des sciences | Faculty of Science
Département de biologie | Department of Biology
Pavillon Gendron | Gendron Hall
30 Marie-Curie Ottawa ON Canada K1N 6N5
☎ 613-562-5718 📠 613-562-5486 bio@uOttawa.ca

Lines 52-53: Done. Text is changed to make it clear that I am pointing out the distinction between qualitative and quantitative evidence of climate change-driven range responses, and focusing on two sets of processes (dispersal and persistence) in particular.

Abstract, line 61: "...climatic conditions elsewhere ...": Where is "elsewhere"? This is not clear. I would remove this part of the statement.

Line 70: "... population ...": Several populations, right? So "populations".

Line 80: No, it was correct as written: "population extinction", but to avoid linguistic ambiguity for some readers, I changed the sentence around so that I refer to extinction rates relative to species and populations.

Line 102: In this paragraph, it would be the best place to define the term dispersal and persistence in more detail. Indeed, to establish in a new location, individuals of a species would need to compete for space and resources.

Lines 115-118: Done.

Lines 108-113: Either remove this section or move it in another section that presents more explicitly the limitations of models.

Done. This text has been moved and modified to expand on the issue of model adequacy, complexity, and simplicity, starting line 329 to 336.

Line 117: "... species sensitivity ..." to be used as "indicator species"? Please clarify and add some references.

Line 127: I have added a synonym from the literature here, and a reference that introduces sensitivity in the context of climate change. This is a concept that is fairly broadly addressed in the climate change impacts literature, so I'm grateful to the reviewer for suggesting that clarification might be needed on this point.

Line 119: "... these techniques...": Sorry, which techniques? Please clarify.

Lines 129-132: I am referring to methods that have been developed and will be reviewed here. I have changed this sentence around to make this clear.

Lines 120-124: This is a long sentence: Please make two or three sentences. Please clarify the last part of the sentence and add some references ("... highly resolved measurements of climate change and its consequent short term variability ...").

uOttawa

Université d'Ottawa | University of Ottawa

Faculté des sciences | Faculty of Science
Département de biologie | Department of Biology
Pavillon Gendron | Gendron Hall
30 Marie-Curie Ottawa ON Canada K1N 6N5
☎ 613-562-5718 📠 613-562-5486 bio@uOttawa.ca

Lines 130-135: Done. This sentence has been broken up and made clearer.

Lines 136-137: “Moreover, many species’ dispersal capacities were thought to be far below rates required to track shifting climatic conditions, ...”. Please add some references.

Line 180: Done. This is still referring to a founding paper that pre-dates nearly all evidence that modern climate change was contributing to range shifts (the reference is from 1986).

Line 162: Please clarify for which species/taxa this shift of 17km/decade referred to.

Line 206: Done. This is a classic meta-analysis from Science in 2011 and it drew on data from many taxa, some of which are listed in this

Lines 211-212: “In some studies, dispersal has been assessed using dispersal kernels.” This sentence can be removed as it is already mentioned.

Done

Lines 218-248: I strongly suggest to only summarize the key processes and take-home message of Leroux et al. 2103 [59] and remove the five equations and respective text. By doing so, the message could be stressing whether or not the modeling approach is applicable to all taxa or not.

*This is an area where correlative (species distribution model) and mechanistic approaches come together. But integrating these approaches is only possible if the mechanisms can be clearly specified. This is a key message of this work, as well as other works that make the case for accounting for critical mechanisms in the context of geographical range shifts. I have expanded the text in this section in many areas (line 277-279, lines 305-313, and the segue to broader issues in lines 329-336) to illustrate how clearly-specified mechanisms are vital *also* because they generate other predictions. There are many changes in this section to make this case as inclusively and accessibly as possible without making the discussion shallow by eliminating its specificity with respect to mechanism. I have also clarified that this section is not about a single paper, as the reviewer seems to suggest, but a body of work*

Lines 249-254: It would be good to cite the book of Kevin Gaston (2003) here (and throughout the text). Gaston KJ. 2003. The structure and dynamics of geographic ranges. Oxford University Press.

Done. Line 311.

Lines 255-261: There are many recent papers that could be relevant to cite here (see below for some suggestions).

uOttawa

Université d'Ottawa | University of Ottawa

Faculté des sciences | Faculty of Science
Département de biologie | Department of Biology
Pavillon Gendron | Gendron Hall
30 Marie-Curie Ottawa ON Canada K1N 6N5
☎ 613-562-5718 📠 613-562-5486 bio@uOttawa.ca

Briscoe NJ, et al. Forecasting species range dynamics with process-explicit models: matching methods to applications. *Ecology Letters*. 2019 Nov;22(11):1940-56.

Getz WM, et al. Making ecological models adequate. *Ecology letters*. 2018 Feb;21(2):153-66.

Naujokaitis-Lewis IR, et al. Uncertainties in coupled species distribution–metapopulation dynamics models for risk assessments under climate change. *Diversity and Distributions*. 2013 May;19(5-6):541-54.

Naujokaitis-Lewis I, Fortin MJ. Spatio-temporal variation of biotic factors underpins contemporary range dynamics of congeners. *Global change biology*. 2016 Mar;22(3):1201-13.

Three of these references are now woven into the text in various places.

Line 319: “Climate chaos”: Any reference for this term?

The first time this term was used in the peer-reviewed literature was actually (as far as I can trace) 1989. I have added that classic reference to the text, line 425.

Lines 330-332: Please remove or expend.

Done. I have modified this text to make it much clearer what I am referring to – and this is an important point: climate (in the way it is formally defined) doesn't harm short-lived organisms. Weather does. Extreme weather is associated with climate change, and the proximate mechanisms for climate impacts will often operate through weather. Line 424-436 addresses this and the problematic sentence on 433-434 has been recrafted for clarity.

Line 350-351: Please add so concluding statement for this sentence/statement.

Line 462-464. Done.

Line 372: Please add some references. These two are only examples of many potential references that could be used from Carlos Carroll and others.

D'Aloia CC, et al. Coupled networks of permanent protected areas and dynamic conservation areas for biodiversity conservation under climate change. *Frontiers in Ecology and Evolution*. 2019 Feb 14;7:27.

Huang JL, et al. Importance of spatio–temporal connectivity to maintain species experiencing range shifts. *Ecography*. 2020 Apr;43(4):591-603.

Done. Line 496.

Line 380: Again, please cite recent reviews like Getz et al. (2018) and Briscoe et al. (2019).

Done – Getz reference inserted. Line 497.

uOttawa

Université d'Ottawa | University of Ottawa

Faculté des sciences | Faculty of Science
Département de biologie | Department of Biology
Pavillon Gendron | Gendron Hall
30 Marie-Curie Ottawa ON Canada K1N 6N5
☎ 613-562-5718 ☎ 613-562-5486 bio@uOttawa.ca

Referee: 2

Comments to the Author(s)
RSPB-2020-2061

Overall comments:

This review covers a timely topic, dispersal and persistence of species in response to climate change. It is well written, and effectively incorporates both empirical observations and models, and tells a story. It will be accessible to a broad audience. It focuses on two recent developments regarding two mechanisms for predicting range shifts related to climate change: 1) dispersal and establishment, and 2) exposure and sensitivity to emerging conditions. Overall, the implications of this field could influence policy and species' persistence. The review develops early ideas of range shifts as singular measurements into being a consequence of multiple anthropogenic drivers (including land use change) that contribute to species redistributions. This then builds to the development of dispersal models in the context of shifting climatic conditions, and the application to conservation for species based on dispersal capacity. There are a range of examples included, from butterflies and birds, to bees and reptiles and more. However, more it focused on terrestrial than marine and freshwater aquatic systems, though the latter are briefly mentioned in passing. It is interesting to think about how aquatic systems deal with climate change, from drying of ponds to warming water, and that in marine systems the oceans are still contiguous unlike terrestrial habitats, therefore some different processes may be operating depending on the species and types of disturbance. The review also includes extreme climate events and its role in extinctions. It presents ongoing challenges, including formulation of models that meaningful but not overly complicated to be so specific and untenable, particularly in the context of applications for conservation.

The figures used are well done and effective in visually communicating main points of the review.

I appreciate the reviewer's comments! I acknowledge the focus on terrestrial systems to a greater extent. I have included a number of references and discuss some examples from freshwater and marine systems. The differences between marine and terrestrial systems is indeed very interesting. I have added a comment on exactly the issue that the reviewer identifies: the relatively small role of factors like habitat fragmentation in systems like coral reefs in comparison with terrestrial environments: Lines 444-446.

Specific comments:

L. 63-64: I don't quite follow this, as it's a jump from talking about species to individual organisms. "...which can be measured on time scales relevant for individual survival and used to inform predictions of local extinction risk."

uOttawa

Université d'Ottawa | University of Ottawa

Faculté des sciences | Faculty of Science
Département de biologie | Department of Biology
Pavillon Gendron | Gendron Hall
30 Marie-Curie Ottawa ON Canada K1N 6N5
☎ 613-562-5718 📠 613-562-5486 bio@uOttawa.ca

Lines 62-64: I have modified this text to make it clear that processes act on individuals, and the net effects can be detected for populations.

L.227: Equation (1): does 'x' have to be defined in this model?

Yes. I really appreciate this catch. x refers to space and that is now stated explicitly on line 281.

L.444-4455: Perhaps "...marine, and freshwater" instead of "...marine, and aquatic"

Done.

Figure 2a: Concerned that the red and green solid lines may not be visible for red/green colourblind readers, although the text does refer to the line position as well.

This is an excellent point – I really appreciate the reviewer's good catch. I have changed the figure with these colours in two ways. First, I have used colours that should not be problematic for the vast majority of readers who may have forms of colour blindness, using a suggested mapping palette from colorbrewer.org and other sources. For those who are completely colour blind, I have also changed the intensity of the colours, so there would still be a shading contrast in the lines on maps. Finally, I have also changed the text in the figure legends for Figure 1 and 2a to point out ways to interpret the maps without referring to line colours. I have changed the colour of Figure 1a also to orange (from green), a colour that most readers should be able to see. This figure did not include any colour contrasts, just a single colour: I am changing the colour for this one simply so that as many readers as possible will see it in a similar way.